# Evaluation of AI-Assisted Telemedicine Service Using a Mobile Pet Application

Sewoong Hwang [1], Yungyeong Song [2] and Jonghyuk Kim [2,*]

1 Graduate School of Information, Yonsei University, 50, Yonsei-ro, Seodaemun-gu, Seoul 03722, Korea; angryking@yonsei.ac.kr
2 Division of Computer Science and Engineering, Sunmoon University, 70, Sunmoon-ro221beon-gil, Tangjeong-myeon, Asan-si 31460, Korea; akag95@sunmoon.ac.kr
* Correspondence: jonghyuk@sunmoon.ac.kr; Tel.: +82-41-530-2266

**Abstract:** This study indirectly verifies the possibility of telemedicine for humans through a mobile application (app) targeting pets. It examined the perception of telemedicine services and the current status of the companion animal industry, the app platform, and its applied technology by industry domain, and four representative types of artificial intelligence (AI) technologies applicable in the medical field. A survey was conducted through an app implementing pet telemedicine, and hypotheses were established and statistically tested based on the adoption period of pets, health status, mobile service utilization (as an index measuring the ease of use of recent AI functions), and positive and negative perceptions of telemedicine services. As revealed by prospect theory, users with a negative perception of pet telemedicine tended to maintain negative perceptions about telemedicine for humans. This study proved that the severity of pet diseases and the ease of use of recent AI technologies act as a moderating effect on the perception of telemedicine services through the verification of reinforcement and additional hypotheses. It suggests a plan to overcome sanctions against telemedicine by utilizing AI technology. A positive effect on changing the medical paradigm to telemedicine and the improvement of the medical legal system were also observed.

**Keywords:** telemedicine; COVID-19; companion animal; application service platform; medical AI technologies; prospect theory

## 1. Introduction

A global debate over telemedicine has begun due to COVID-19. The Washington Post has addressed Forrester Research's view that the number of telemedicine uses, including treatment for COVID-19, will increase to about 1 billion by the end of 2020 [1]. In addition, IHS Markit, a global information provider in the United Kingdom, forecasts that the global demand for telemedicine will reach 150 million in 2022 from 39 million in 2018 [2]. While the problem of shortage of medical institutions around the world is on the rise, the annual demand for telemedicine—particularly in North America and Asia—is increasing rapidly due to such shortage problems. According to a 2019 digital health survey conducted by Accenture, nearly half of healthy young Americans aged between 22 years old and 38 years old only use routine healthcare services. They want medical services that are more convenient and easier to access than the quality healthcare enjoyed by the previous generation. For example, they hope to have easy access to medical services through online or mobile healthcare applications (apps) [3]. In the United States, where telemedicine was common before COVID-19, plans are currently underway to implement it as an easy alternative to treating several mild diseases. In other words, by using a smartphone or a computer equipped with a webcam, patients can communicate with doctors 24 h a day, 7 days a week, and receive high-quality medical support and treatment at home through national medical communities or local medical programs, in addition to their potential in being used in preparation for the outbreak of a new virus, such as the recent SARS-CoV-2.

The COVID-19 pandemic situation has provided an opportunity to avoid primary face-to-face contact between patients and medical staff [4]. This preventive action through digital telemedicine has the advantage of maximizing the treatment effect by minimizing contact between patients with mild symptoms and severe patients with underlying diseases [5]. The current status of introducing telemedicine in major developed countries that have adopted telemedicine early is summarized in Table 1.

**Table 1.** Introduction of telemedicine in major countries.

| Nation | Status | Resource |
|---|---|---|
| The US | - 1 out of 6 outpatient medical cases remotely conducted (as of 2019 after telemedicine implemented in 2014)<br>- More than 80% of patients wish to obtain a prescription through telemedicine, receiving regular medications<br>- 25% of outpatients of young adults in their 20 s to 40 s use telemedicine in 2019 | CDC (Center for Disease Control and Prevention) |
| France | - Completed a preliminary feasibility review for telemedicine for 3 years after 2016<br>- Legislated telemedicine from 2019 and began implementing nationwide<br>- Telemedicine is recommended first if a doctor determines that a patient's condition is not serious | Santé publique France (Public Health France) |
| Germany | - Since 2018, telemedicine is legally allowed for some mild patients<br>- Began to legalize the world's first patient prescription using an application | RKI (Robert Koch Institute) |
| Japan | - Expanded the plan to use telemedicine for the first visit with COVID-19 as a trigger<br>- Strives to establish legislation to continuously expand the scope of telemedicine | JPHA (Japan Society of Public Health) |
| China | - Started telemedicine in 2014 on the initiative of the government<br>- Currently recorded the fastest growth rate among Asian countries<br>- Operated more than 300 online hospitals nationwide | Chinese CDC |
| Chile | - In many Latin American countries, telemedicine has been partially allowed and is being implemented since early 2012 to cope with AIDS patients.<br>- Since 2017, Chile has been allowing telemedicine for severely ill patients such as cancer and elderly patients<br>- However, legalization or medical system cannot support the demand of private patients and medical staff | National Institute of Public Health of Chile (ISPCH—Instituto de Salud Pública de Chile) |

As shown in Table 1, telemedicine is already quite pervasive in major developed countries, with most countries making efforts to allow telemedicine through legislation after COVID-19. In contrast, many other countries, including Korea, are still opposed to telemedicine because of the safety and responsibility issues in terms of patients or the general public directly reporting their condition to doctors and receiving appropriate

diagnoses and prescriptions. In the case of Korea, although telemedicine legislation was announced twice in 2010 and 2014, it fell through due to opposition from the medical association. The proposed telemedicine plan remained pending at the National Assembly Health and Welfare Committee but was finally discarded when the schedule of the National Assembly was ended [6]. Despite having excellent information and communications technology (ITC) and medical staff, the potential competitiveness of telemedicine has not been realized due to policy restrictions. This can be attributed to the relatively high intensity of medical regulations in Korea, which are also closely related to policies aimed at protecting the rights of medical associations [7]. The American Telemedicine Association (ATA) has determined through several studies that telemedicine is necessary due to an increase in aging and chronic diseases such as high blood pressure and diabetes. Despite this necessity, the issue of the practical expansion difficulty due to policy reasons of various interest groups continues to be raised [8]. Although the interest in telemedicine is increasing due to COVID-19, it is difficult to actually introduce it and discuss the timing in detail because it is directly related to the health of people and involves consultations with various interest groups. However, with the emergence of COVID-19, young people, who have a relatively low need for medical care, and those in the medical blind spot strongly argue that telemedicine is necessary for the public [9]. Therefore, we considered the problem of telemedicine caused by COVID-19 and decided that it was necessary to first study the introduction of telemedicine for companion animals prior to the introduction of telemedicine for humans. Korean society alone has over 15 million adopted companion animals; as the number of adopted companion animals increases exponentially around the world, related research and opportunities for data acquisition also increase. This environment has also given rise to various telemedicine mobile app services for pets. We use this to investigate people's perceptions about pet telemedicine services and predict the introduction of telemedicine for humans in the future.

This study comprises four sections as shown in Figure 1. Section 2 will review studies on the types and utilization of platforms through online or mobile apps by industry domain. In addition, we try to infer the trend of changes in people's perception about the use of information technology (IT) for pet telemedicine by reviewing past research on the companion animal industry. Finally, we attempt to understand the procedural agreement between the government and interest groups through past research on the introduction of telemedicine services to solve conflicts among them. Based on these preceding studies, we form the analysis constructs of cognitive change that can cope with the smooth introduction of telemedicine for people in the future and establish the hypotheses of the model. In Section 3, we perform statistical analyses based on actual users of pet telemedicine services. We will conduct quantitative analysis using statistical techniques to establish and verify the hypotheses, in addition to descriptive statistics. In Section 4, we evaluate the academic contribution to the theory verified by actual data and provide a summary of the whole study. Finally, we will conclude this study by revealing the implications, limitations, and future research plans.

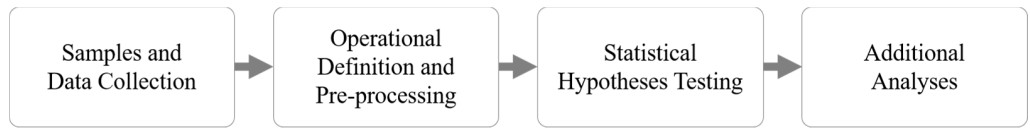

**Figure 1.** Research steps.

## 2. Background

### 2.1. Application Service Platform and Its Technologies

The Fourth Industrial Revolution has led to changes in various aspects of our lives, among which the hyper-connectivity realized by platform technology is significantly influencing social operating entities by producing and sharing data from multiple objects [10]. In addition, in the field of personal creation with few hurdles to entry, many individuals act

as streamers—creating, distributing, and sharing content through a platform, and becoming economic actors who earn their own profits along with the development of the platform industry [11]. Each industry area is developing a future-oriented platform that meets the needs of the industry domain by applying various artificial intelligence (AI) technologies and methodologies. This creates services through web services and mobile apps that can be used anywhere through a simple user interface (UI). In Table 2, we categorize and summarize platform technologies that have already been commercialized in many fields or are presented in various methodologies through related research.

**Table 2.** Online and mobile platforms by industry domain.

| Nation | Resource |
|---|---|
| Tourism and Culture | - Google Art & Culture (Virtual Reality Tech.)<br>- Cyber Literature Plaza (Virtualization)<br>- Shared accommodation platform and traveler customized platform (Personal Customized Suggest System) |
| Education | - Education platform for the real estate industry (Sharing Cloud)<br>- Convergence education platform for cultivating convergent talents (Deep Bidirectional Transforming Tech.)<br>- E-learning learning support platform through real-time two-way communication (Natural Language Process)<br>- Open source hardware-based IoT education platform (Internet of Thing) |
| Finance | - Financial ARS (Voice Recognition) using AI<br>- P2P loan brokerage platform (P2P)<br>- Block-chain-based financial platform (Distributed Accounting Tech.)<br>- Digital insurance platform service (Chatbot Tech.) |
| Retail and Distribution | - Big data platform for the digital transformation of agriculture and rural areas (Digital Transformation Tech.)<br>- Online open market platform (GIS and CTI)<br>- Web cartoon platform (Deep Learning)<br>- Fashion Business Platform (Customized Suggest System)<br>- Platform transportation company (Logistic Optimization) |
| Manufacturing | - Manufacturing data analysis platform (Real-time Detection)<br>- Manufacturing-specific library-based big data analysis platform (Cloud-based Program)<br>- Smart manufacturing big data platform for predictive diagnosis of manufacturing robot failure (Real-time Modeling) |
| Medical | - U-Care (Customized Monitoring System) for the health and safety of the elderly in a single household<br>- A community for responding to infectious diseases based on information and communication technologies (ICT) (Robotic Sensing Modeling)<br>- IoT-based intensive care unit real-time monitoring platform (Real-time Monitoring Tech.)<br>- Consultation on the use of ICT between doctors and medical personnel for the elderly (Telemedicine System)<br>- Convergence platform for personalized companion animal service (Open Collaboration Tech.) |

The status and methodology of each field classified in the table above are as follows: First, the tourism and cultural industry is a representative platform app field that has been widely used by consumers by developing shared accommodation apps such as Airbnb from an early stage. According to a study on the properties of a shared accommodation platform, consumers want to use such platforms because of four factors:

1. the economic feasibility of saving money and time;
2. the aesthetics that emphasized the external part;

3. the informational aspect that enables perceptions about a product through content provided by the platform;
4. the convenience of a platform, which acts as an intermediary so that suppliers and consumers can remove restrictions on time and space [12].

These platforms have a fairly positive effect on the product and help increase consumers' desire to repurchase [13]. The development of the platform is also related to personalized recommendation technology. Segmentation-based mass-production customized strategies were used in the past, while recent AI-based recommendation technology has had a significant impact on the tourism and cultural industries. An online app related to theme travel guides that have appeared recently, together with certified experts, proposes matching services to consumers who want personalized travel such as medical tourism or special-purpose business trips [14]. Launched in 2011, Google Arts & Culture is a platform that helps provide appreciation and commentary on artworks through virtual reality, sharing cultural and art content from around the world [15]. In terms of education, it presents the latest platform in the era of the Fourth Industrial Revolution.

Education system software can use the cloud environment Smart Space EduPlatform (SSEP) to foster talent for the real estate industry [16]. Another platform presents operational plans to consumers through customized online-to-offline (O2O) strategies that can be selected by individuals in order to cultivate talents who can adapt to various convergence environments [17]. With regard to finance, investment in and utilization of the platform industry has already entered a stable phase. Most financial companies are mass-producing AI-based platforms by mobilizing strong financial power. Along with a study verifying the effectiveness of a chatbot system for the promotion of insurance and financial products while providing customized services to individual customers, there is also a study on digital insurance platform services using Fintech and Insuretech (a compound word made by joining "insurance" and "technology") [18,19]. In addition, there is also research on security-related platforms such as adding simple authentication and personal information to increase the convenience of financial products to customers [20,21]. On the distribution side, there is an example of developing a prototype of a personal clothing custom platform to meet the needs of consumers. In other words, consumers can choose clothes through the design process, while a system for recommending designs of products based on asking questions and receiving responses about their desired style is simultaneously operational. Through this, the needs of consumers can be immediately reflected [22]. In Korea, online shopping malls in the form of intermediary platforms have developed into social commerce companies such as 11st Street, G-Market, and Auction. In addition, food delivery-related apps that are enjoying the best boom due to the recent COVID-19 outbreak can be said to be a platform suitable for the "untact" (non-contact) era [23]. With regard to manufacturing, there is a manufacturing-specific library platform that allows non-professionals to analyze data and solve problems; this model has been applied to various manufacturers through many studies, proving its effectiveness and validity [24]. In addition, open source, a robot technology equipped with a vibration sensor, was developed and applied to manufacturing. In recent years, there have been an increasing number of cases of establishing a fault diagnosis platform that informs analysis results in real-time through a cloud system that collects, transmits, and manages data [25].

The use of these platforms in the medical field can be captured as follows. First, the Internet of Things (IoT)-based intensive care unit real-time monitoring platform for humans can collect data from sensors attached to patients through an IoT bridge and deliver information to medical workers. This facilitates patient monitoring and helps minimize medical gaps [26]. As a case study that can be applied to animals, to prevent economic loss due to infectious diseases, information and communication technology (ICT) is used to develop a remote system so that farms or livestock can be managed. In addition, it is possible to record the body temperature of livestock with a temperature sensor attached to each animal and transmit it to a farmer or a veterinarian remotely through a mobile app service [27].

## 2.2. Advanced Technology Based on AI

Machine Learning has made great advances in pharmaceuticals and biotech efficiency. With mHealth technology, patients can stay connected to their doctors to monitor infectious diseases such as COVID-19. Based on video processing, diagnosis is also possible without a contact device [28]. According to prior research, AI technologies applicable in the medical field can be classified into four technologies as shown in Figure 2.

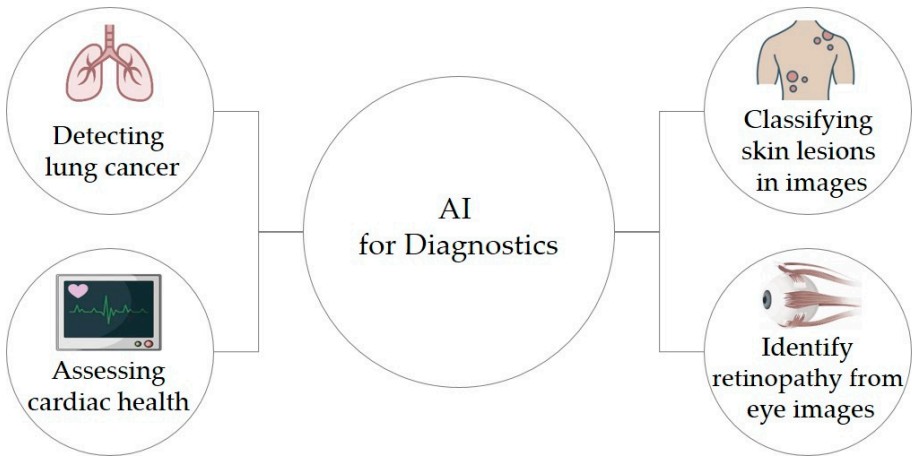

**Figure 2.** Artificial intelligence (AI) for diagnostics.

The first is AI for diagnostics. Diagnosing diseases takes years of medical training. This technology is time-consuming, putting doctors under strain and often delaying life-saving patient diagnostics [29]. Machine learning, especially deep learning algorithms, have recently made huge advances in automatically diagnosing diseases, making diagnostics cheaper and more accessible [30]. These algorithms can learn to spot patterns similar to the way doctors see them. A key difference is that algorithms need a lot of concrete examples that need to be neatly digitized, such as detecting lung cancer or strokes based on computed tomography (CT) scans, assessing the risk of sudden cardiac death or other heart diseases based on electrocardiograms and magnetic resonance imaging (MRI) images, classifying skin lesions in skin images, and identifying indicators of diabetic retinopathy in eye images. Since there are plenty of good data available in these cases, algorithms are becoming just as good at diagnostics as the experts. The difference is that the algorithm can draw conclusions in a fraction of a second, and it can be reproduced inexpensively across the world [31,32]. The second classification is the individualization of treatment. Different patients respond to drugs and treatment schedules differently. Therefore, individualized treatment has enormous potential to increase patients' life spans. However, it is very hard to identify which factors should affect the choice of treatment [33]. AI can automate this complicated statistical task and help discover the characteristics that indicate or predict that a patient will have a particular response to a particular treatment. AI algorithm-embedded systems cross-reference similar patients and compare their treatments and outcomes. The resulting outcome predictions make it much easier for doctors to design the right treatment plan [34]. The third classification is AI in medical imaging, especially, imaging devices using a digital twin. A digital twin is a computer-modeled replica of a physical person. In healthcare, digital twins can be used to model both medical devices and patients to see how the former work on people with specific conditions as types of virtual images. Disease detection, classification, and quantitative assessment in medical imaging are important for early diagnosis and treatment planning [35]. Several techniques have been proposed for segmenting medical image data through quantitative assessment. However, some quantitative methods of evaluating medical images are inaccurate and require considerable computation time to analyze large amounts of data. Analytical methods using AI algorithms can improve diagnostic accuracy and save time [36]. The

fourth and final classification is AI as a telemedicine-assisted function. The increase of telemedicine—the remote treatment of patients—reflects the rise of possible AI applications. AI can assist in caring for patients remotely by monitoring their information through sensors. A wearable device may allow for constant monitoring of a patient and the ability to notice changes that may be less distinguishable by humans. The information can be compared to other data that has already been collected using AI algorithms that alert physicians on any issues. However, there are some issues about the limitations of monitoring in order to respect a person's privacy. Nevertheless, it is true that AI algorithms are essential for telemedicine activation in the COVID-19 era [37,38]. However, telemedicine still has some technical limitations. First of all, when a doctor diagnoses a patient remotely, it is difficult to recognize the psychological state. Moreover, the possibility of false diagnosis cannot be ignored depending on the performance of the mobile camera or image recognition algorithm. In addition, there is a limit to detailed diagnosis with only basic mobile devices. The patient's blood pressure and pulse can be diagnosed with a simple device, but it is difficult to know the nerve response or internal state of the body [39,40].

### 2.3. Industry Review of Companion Animals

In Korea, the companion animal population exceeds 15 million. According to the Ministry of Agriculture, Food, and Rural Affairs, the percentage of households with companion animals is 26.4% as of 2019. This means that one out of four Korean households have adopted companion animals. Recently, a new word called PETFAM, a compound word made by joining "pet" and "family," has emerged, indicating the homogeneity of households with companion animals. Figure 3 shows the increasing trend of trademark applications related to companion animals. The growth rate for the past five years has exceeded the annual average of 12%, and the types are also diverse, varying from funeral, training, beauty, lodging, and hotel businesses. Euromonitor, a global market research firm, announced its performance in the companion animal care market and predicted that it would generate sales of USD (US dollars) 139.8 billion by the end of 2020 [41]. According to the latest household trend survey data by the Korea National Statistical Office, the size of the companion animal market has nearly tripled from USD 400 million to USD 1.2 billion in just a decade from 2006 to 2016. The total market size is expected to increase to USD 35 billion by 2027 [42].

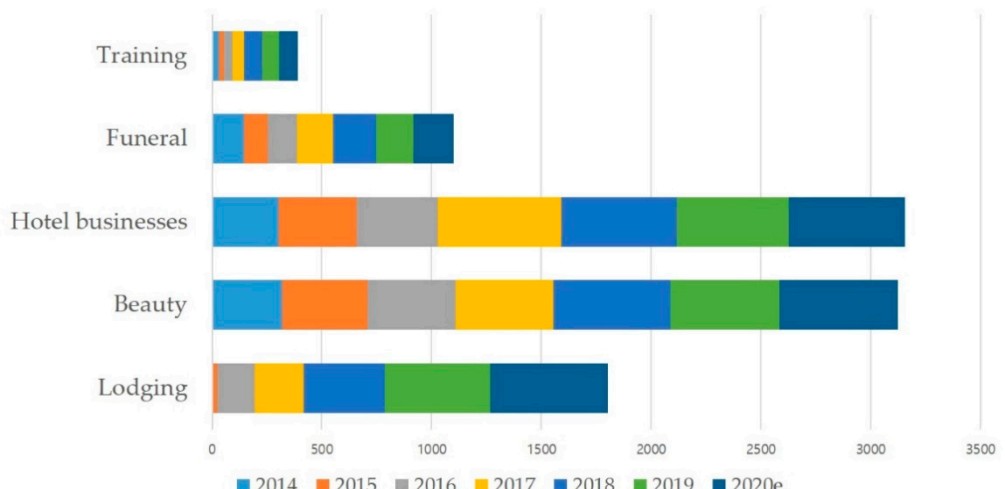

**Figure 3.** Number of applications by business type related to companion animals.

While the companion animal industry is steadily increasing, caregivers have demanded improvements in the legal system. Companion animals relate to people's right to seek happiness under the constitution; therefore, the rights of the owner of companion animals can be treated more exceptionally in terms of protection and welfare than owners

of other animals such as ordinary mammals, birds, and reptiles; moreover, the strength of the relationship can even be interpreted differently [43]. In particular, in the case of Korea, companion animals do not have a special medical system except for the Veterinary Law and the Pharmacist Law. Due to the current laws and regulations, medical staff are not obligated to write and issue companion animal diagnosis results, which may lead to over-treatment problems. First of all, there is virtually no remedy for injuries or death during the treatment of companion animals [44]. Medical expenses are also not limited by law, and hence medical and surgery expenses are different for each hospital [45]. In Germany, where medical expenses are limited to within a certain amount of the legal fee, a system that relieves the burden of caregivers is urgently needed [46]. With regard to the registration and payment of pet ownership tax, which are currently major issues in various countries, it can be expected that they are resolved with positive synergy effects by referring to examples of companion animals. In Germany, for example, the pet ownership tax is used as a cost for cleaning pet waste or for operating a dog shelter [47].

### 2.4. Perception of Telemedicine Services

Telemedicine service refers to the provision of medical services in a non-face-to-face format by medical staff using images or other remote equipment for patients as IT technology develops. It also refers to all services that receive doctors' opinions through phone, text, or remotely, in addition to video chat, monitoring patients, and transmitting physiological data through wearable devices [4]. If telemedicine is generally allowed, collaboration with overseas medical staff is possible, which can provide an opportunity to ensure the continuity of medical services. Telemedicine services are being implemented or considered in the United States, China, Japan, and many European countries. These countries have also introduced AI technology to enable more precise treatment. In particular, telemedicine services in China, which has a large population and vast territory, are estimated to have over 100 million users based on easy accessibility [48]. Telemedicine services have the advantage of ensuring the health of inaccessible local residents, increasing convenience, and fostering new industries related to medical care. Conversely, there are also disadvantages. Patients may be concentrated in certain famous large hospitals, and there are concerns about the privatization of medical care and leaking patients' personal information [49]. However, in recent years, in a global pandemic environment such as that brought about by COVID-19, a shift in perception about the non-face-to-face treatment being needed to protect the health rights of the people and to keep both medical staff and patients safe from infectious diseases has begun to occur [5]. Many Korean studies on telemedicine deal with conflicts between the government and medical associations. Due to COVID-19, the Ministry of Health and Welfare allowed non-face-to-face treatment, such as video telemedicine and remote monitoring, for viral infections from 24 February 2020. However, the medical association maintains the position that it cannot be allowed because technical stability and legality are not guaranteed. In addition, their opposition cites the possible collapse of the healthcare infrastructure, which could worsen hospital profitability. Their arguments are that, while treatment may be possible remotely, the actual effectiveness of telemedicine will be reduced because medications require direct visits to the pharmacy to fill prescriptions. Similarly, they argue that it is not a prudent decision to simply allow telemedicine without legally solving many problems such as leaking sensitive personal information, the responsibility for medical accidents, and the application of medical insurance premiums [50]. For this reason, despite the fact that telemedicine is being implemented around the world, in the case of Korea, there remains a situation in which the first step has not been taken due to safety issues and opposition from medical groups. A Korean survey on the perceptions about telemedicine by nursing students, nurses, and the general public reveals that nursing students and the general public have more positive perceptions about telemedicine than do nurses. While nurses positively evaluated the efficiency of time, they perceived the safety of telemedicine to be low in aspects that are directly related to the patient's life [51]. Table 3 summarizes the general confrontational claims of the government and medical as-

sociations on sensitive matters such as telemedicine and the composition of working-level consultative bodies, which have recently become an issue due to COVID-19.

**Table 3.** Comparison of telemedicine between the government and medical associations.

| Issues | Pros (Representing the Gov.) | | Cons (Representing the Medical Ass.) | |
|---|---|---|---|---|
| Telemedicine | - | Expected synergy effect by combining IT technology and health care service | - | Possible to raise safety issues such as risk of misdiagnosis and unfaithful medical treatment |
| | - | Alleviate patient distraction and demand for large general hospitals | - | Threats to the survival of small-scale local hospitals due to lack of demand |
| Incorporation of a Medical Institution | - | No change in the governance structure of hospitals, a representative non-profit corporation | - | Acceleration of commercialization of the parent hospital, due to the concentration of subsidiary commercial businesses |
| | - | Institutional deregulation is required for hospital exports and increase of foreign patients | - | Possible collapse of the public medical system as a preliminary stage for practical medical privatization |
| | - | Not complete privatization of public medical facilities | | |
| Low Medical Rate Problem | - | Agree to increase the medical fee within an appropriate range | - | As the public medical fee is formed as low as about a quarter or half of the standard fee, possible to spread abnormal systems such as uninsured or excessive medical care |
| | - | However, the increase must be adjusted within the health insurance finances | - | Urgent need for realization or fundamental improvement of the health insurance system |
| Organization of Council for System Improvement | - | Possible to propose a council in which the medical community and national representatives participate at the same time (headed by the Minister of Welfare) | - | Establishment of a special committee for medical reform under the direct control of the President (or Prime Minister) and request for legalization as an independent organization |

The current status of telemedicine for companion animals in Korea is as follows: Several studies have developed platforms for companion animals and, through research, indirectly verified the effectiveness of telemedicine for humans. The POSCO Research Institute (POSRI) is leading this research [52,53]. On the other hand, veterinary-related organizations are in opposition. According to the Korean Veterinary Law or veterinary hospital facility standard draft, telemedicine by veterinarians is not permitted in principle, and a series of treatment activities, such as treating animals and dispensing or administering medicines, must be performed by veterinarians [54]. Nevertheless, several veterinary hospitals are providing services to check the condition of animals and provide counseling through online or mobile social network services. They even issue prescriptions according to the condition of animals through e-mail. As such, cases of consulting and treatment using IT technology are steadily increasing. On the surface, mobile app services in Korea also advocate legitimate services such as pet supplies sales, veterinary hospital price comparisons, and healthcare solutions, but the service they actually provide through the app is telemedicine. Some Korean application services are "Shiny," "WAAL," "IntoPet," "FITPET," "MyPetPlus," "Pet Price," and "Ajinyangi." These mobile app services actually involve consultations with veterinarians who even provide prescriptions. We thus conducted a survey on real users to study the public's perceptions about mobile app services regarding

telemedicine for companion animals currently being used. The purpose of this study is to predict the situation in which telemedicine services for humans will be introduced in the future and to suggest the possibility of treatment and service level.

### 3. Methods

#### 3.1. Samples and Data Collection

In this study, we collected research data through "OnPanel," a mobile survey research organization. The survey respondents were 2900 OnPanel members who adopted and cared for companion animals. Because the survey contents of 44 respondents were unreliable and analytically inappropriate, we used the data of 2856 respondents as the final analysis sample. We used the UI of the mobile app "FITPET" to measure the ease of application among the surveyed users.

As shown in Table 4, there were 1643 females (57.5%) and 1213 males (42.5%). In terms of the age groups, 1098 were under the age of 30 (38.4%) years old, while 409 were elderly people over the age of 50 (14.3%) years old. In terms of the adoption period, 920 people (32.2%) had adopted for less than one year or more than seven years. With regard to the health status, 1782 respondents (62.4%) said their pets were in good health, while 1074 (37.6%) answered that a certain degree of medical treatment was necessary. Among them, 291 (27.1%) of the respondents answered that they own a pet with a serious disease, while 783 (72.9%) had pets with other mild diseases.

**Table 4.** Descriptive data summary.

| Category | | Frequency | % |
|---|---|---|---|
| Gender | Female | 1643 | 57.5 |
| | Male | 1213 | 42.5 |
| Age | Under 30's | 1098 | 38.4 |
| | 30–40's | 786 | 27.5 |
| | 40–50's | 563 | 19.7 |
| | 50–60's | 282 | 9.9 |
| | Over 60's | 127 | 4.4 |
| Adoption Period | Under 1 year | 439 | 15.4 |
| | 1–3 years | 665 | 23.3 |
| | 3–5 years | 618 | 21.6 |
| | 5–7 years | 653 | 22.9 |
| | Over 7 years | 481 | 16.8 |
| Health Condition | Good or no problem | 1782 | 62.4 |
| | Need medical Treatment | 1074 | 37.6 |
| Total | | 2856 | 100.0 |

#### 3.2. Operational Definition and Pre-Processing

We performed pre-processed and determined basic demographic information through the survey data and performed the hypothesis establishment and verification sequentially. We conducted a quantitative test using SAS University edition (100 SAS Campus Drive Cary, NC 27513-2414, USA), a free-source statistical package with Oracle Virtual-Box, a virtual machine (500 Oracle Parkway Redwood Shores, CA, 94065, USA), specifically used for testing the basic hypotheses. We also used partial least squares (PLS), one of the distribution-based techniques, because we sought to explain the relationship between each construct. PLS is relatively independent of the assumptions about the normal distribution of variables and residuals and mainly uses a method of minimizing measurement errors to increase the accuracy of path coefficients. This process is an attempt to reproduce the

observed covariance matrix using the maximum likelihood function [55]. We estimated the model using Smart-PLS 3.0 for the robustness analysis, an additional hypotheses test. The moderation effect was measured with the Z-score obtained through Cohen's equation and the Sobel test [56]. In order to analyze the reliability, convergent validity, and discriminant validity of measurement items (constructs), we analyzed all loading values related to the variables of the measurement model; items that did not satisfy 5% of the significance criteria were excluded from the analysis targets [57].

First, as shown in Table 5, we applied a factor loading criterion of 0.6 or higher for each questionnaire to secure the reliability of the measurement items [58]. It can be argued that the reliability of individual items was thus guaranteed because all items met the criteria. Second, we measured the composite reliability (CR) and Cronbach's alpha values of each construct. As a result, all critical values were above 0.7 and the average variance extracted (AVE) was also above 0.5; thus, the validity of internal consistency was secured.

**Table 5.** Reliability and convergent validity.

| Variables | | Items of Measurement | Factor Loading | CR | AVE | CRB Alpha |
|---|---|---|---|---|---|---|
| Ease of use of mobile application services | Q1 | Anyone can easily download the mobile app service | 0.794 | 0.924 | 0.668 | 0.847 |
| | Q2 | App service menu is easy for user to select | 0.786 | | | |
| | Q3 | App service can easily reach the desired information | 0.763 | | | |
| | Q4 | The frame for uploading information is concise | 0.827 | | | |
| | Q5 | User-friendly process flow | 0.813 | | | |
| Positive perception of pet telemedicine service | Q6 | Pet remote counseling service (for professionals) is useful | 0.736 | 0.836 | 0.564 | 0.774 |
| | Q7 | Pet remote consultation service (general inquiry) is useful | 0.764 | | | |
| | Q8 | Pet telemedicine services are useful | 0.859 | | | |
| | Q10 | Pet remote prescription service is useful | 0.763 | | | |
| Positive perception of telemedicine service for people | Q16 | Remote consultation service for people (for professionals) would be useful | 0.863 | 0.887 | 0.639 | 0.822 |
| | Q17 | Remote consultation service for people (general inquiries) would be useful | 0.841 | | | |
| | Q18 | Telemedicine services for people would be useful | 0.854 | | | |
| | Q20 | Remote prescription services for people would be useful | 0.821 | | | |

**Table 5.** *Cont.*

| Variables | | Items of Measurement | Factor Loading | CR | AVE | CRB Alpha |
|---|---|---|---|---|---|---|
| Negative perception of pet telemedicine service | Q11 | Pet remote counseling service currently in operation is illegal business | 0.767 | 0.792 | 0.549 | 0.764 |
| | Q13 | Pet telemedicine service currently in operation is illegal business | 0.761 | | | |
| | Q15 | Pet remote prescription service will be illegal | 0.816 | | | |
| Negative perception of telemedicine service for people | Q21 | Remote consultation service for people (for professionals) would be illegal | 0.674 | 0.876 | 0.630 | 0.821 |
| | Q22 | Remote consultation service for people (general inquiries) would be illegal | 0.679 | | | |
| | Q23 | Telemedicine services for people would be illegal | 0.693 | | | |
| | Q25 | Remote prescription services for humans would be illegal | 0.724 | | | |

Third, the criterion for accepting discriminant validity is that the square root of a latent variable should generally be greater than the correlation with other latent variables [59]. As shown in Table 6, the square root of AVE, which is the diagonal value of each recursive variable located on the diagonal, was higher than the correlation value of the same row or column; thus, the discrimination validity was secured. On the other hand, the correlation between all variables was less than 0.8; therefore, there was seemingly no problem of multicollinearity between variables [60]. Moreover, with regard to the multicollinearity, the value of the variance inflation factor (VIF) was found to be less than 1.3, which can be interpreted as having no problem of multicollinearity between the latent variables. However, since all variables included in the PLS model have their own modified and supplemented aspects to fit the context of this study, even though they were designed by previous studies, there may be a common method bias. This occurs when the independent and dependent variables are measured using the same measurement tool and survey respondents. To minimize this problem, we structured the content of the survey so that it was difficult for survey respondents to guess the surveyor's intention and its outcome. In addition, we performed Harman's single factor test to test general method bias after investigation [61].

**Table 6.** Discriminant validity.

| Constructs | (A) | (B) | (C) | (D) | (E) | VIF |
|---|---|---|---|---|---|---|
| Ease of use of mobile application services (A) | 0.812 | | | | | 0.572 |
| Positive perception of pet telemedicine service (B) | 0.743 | 0.812 | | | | 1.082 |
| Positive perception of telemedicine service for people (C) | 0.237 | 0.741 | 0.793 | | | 0.982 |
| Negative perception of pet telemedicine service (D) | 0.548 | −0.571 | −0.223 | 0.834 | | 1.106 |
| Negative perception of telemedicine service for people (E) | 0.348 | −0.314 | −0.585 | 0.642 | 0.864 | 1.067 |

*3.3. Hypotheses*

To establish the hypothesis of this study, we first considered the relationship with the adoption period of companion animals. In other words, we believed that people who have been with animals for a long time will be more positive in their perceptions of telemedicine services. Therefore, we set the first hypothesis as follows:

**Hypothesis 1 (H1).** *Positive perceptions about pet telemedicine services differ depending on the period of adoption of companion animals.*

In addition, we hypothesized that people's perceptions of telemedicine services will differ according to the health status of companion animals. In other words, it is highly likely that the owners of healthy animals did not seriously consider telemedicine services. Therefore, the second hypothesis was set as follows:

**Hypothesis 2 (H2).** *Positive perceptions about pet telemedicine services differ depending on the health status of the companion animal.*

In addition, in order to establish a hypothesis on the usability of the technology, we tried to examine the difference in perceptions of telemedicine services according to the degree of ease of using mobile apps, which varies from one person to another. Moreover, we set the third hypothesis by subdividing it in order to examine the difference in perceptions of both pet and human telemedicine services.

**Hypothesis 3a (H3a).** *Positive perceptions about pet telemedicine services differ depending on the degree of ease of use of mobile apps.*

**Hypothesis 3b (H3b).** *Positive perceptions about telemedicine services for people differ depending on the degree of ease of use of mobile apps.*

Finally, we divided the perception of telemedicine services into groups with positive and negative tendencies. We set the following hypotheses to evaluate the positive and negative tendencies for pet and human telemedicine services.

**Hypothesis 4a (H4a).** *People with positive perceptions about pet telemedicine services also have positive perceptions about telemedicine services for people.*

**Hypothesis 4b (H4b).** *People with negative perceptions about pet telemedicine services also have negative perceptions about telemedicine services for people.*

## 4. Results

*4.1. Statistical Hypotheses Testing*

The results of the hypotheses tests were determined by referring to the path coefficient values and t-values for each hypothesis.

Table 7 presents the results of the analysis. With regard to H1, we concluded that there was no statistically significant relationship between the length of adoption of companion animals and the degree of perception of telemedicine services. In other words, even if the companion animal has been with the user for a long time, there is no direct significance of the user needing a telemedicine service for the companion animal. On the other hand, as shown in H2, depending on the health status of the companion animal, the level of perception of pet telemedicine services is significantly different. That is, when the companion animal is not in good health, the user's perception of the telemedicine service is positive compared to when the companion animal is in good health. In the case of H3, it was statistically verified that the ease of use of mobile app services has a positive effect on the perception of telemedicine services for companion animals and people. In other

words, those who find it easy to use the latest technology had more positive perceptions about telemedicine using IT technology. H4 is a hypothesis relating to positive–negative evaluation and, in summary, it seeks to determine whether users with positive or negative perceptions about telemedicine for companion animals have the same perceptions about telemedicine for people. The results revealed that users with positive perceptions of pet telemedicine did not maintain positive perceptions about telemedicine for people. Conversely, users with negative perceptions toward telemedicine for animals had a high tendency of maintaining these negative perceptions even about telemedicine for people. This is in line with the results of Tversky and Kahneman's prospect theory [62]. In other words, dissatisfaction with negative perceptions feels much more ineffective than does satisfaction with positive perceptions of the same intensity. This study shows that the concept transfer of negative recognition was more effective than that of positive recognition.

**Table 7.** Result of hypotheses test with path coefficient.

| Hypothesis | Path | Path Coefficient | t-Value | Test Result |
|:---:|:---:|:---:|:---:|:---:|
| H1 | Adoption period → positive perception of pet telemedicine service | 0.108 | 0.027 | Reject |
| H2 | Health condition → positive perception of pet telemedicine service | −0.527 | 6.507 *** | Accept |
| H3a | Ease of use of mobile application service → positive perception of pet telemedicine service | 0.484 | 5.148 *** | Accept |
| H3b | Ease of use of mobile application service → positive perception of telemedicine service for people | 0.371 | 2.986 ** | Accept |
| H4a | Positive perception of pet telemedicine service → positive perception of telemedicine service for people | 0.065 | 0.211 | Reject |
| H4b | Negative perception of pet telemedicine service → negative perception of telemedicine service for people | 0.413 | 3.492 ** | Accept |

(Significant Level: ** $p < 0.05$, *** $p < 0.01$).

### 4.2. Additional Analyses (Moderating Effect and Robustness Check)

We tried to analyze the moderating effect on H2, H3, and H4b for in-depth consideration. We sought to conduct additional analyses through a robustness test to verify the outcomes of the basic hypotheses. Thus, we set up additional hypotheses as follows. Since H3a and H3b showed significant results, we set the reinforcement hypotheses H5a and H5b in order to verify them again. In addition, since H2 showed a significant result, we determined that it was necessary to further subdivide and examine the health status of companion animals; thus, H6 was set and tested. We also set H7 to examine the specific effects of the degree of ease of use of mobile app services on the negative effects of telemedicine further, as H4b appeared to be a significant result.

**Hypothesis 5a (H5a).** *Negative perceptions about pet telemedicine services differ depending on the degree of ease of use of mobile app services.*

**Hypothesis 5b (H5b).** *Negative perceptions about telemedicine services for people differ depending on the degree of ease of use of mobile app services.*

**Hypothesis 6 (H6).** *The severity of a companion animal's disease has different effects on the companion animal's health status and positive perceptions about pet telemedicine services.*

**Hypothesis 7 (H7).** *The degree of ease of use of mobile app services has different effects on the negative perceptions about pet and human telemedicine services.*

## 5. Discussion

From the results of H5 in Table 8, we confirmed that the ease of use of the app has a positive effect on telemedicine and a negative significant effect on negative perceptions. In other words, among users who are highly adept at using apps, negative perceptions about telemedicine for pets and humans are reduced. We were thus able to reinforce the significance of the existing hypothesis. In addition, according to the results of H6, we confirmed that interest in telemedicine is rather low when the companion animal's disease is severe. In other words, H2 reveals that users have a high positive perception of telemedicine in case of poor health. However, it was confirmed that the H6 test by moderation effect played a role in lowering interest in telemedicine in severe cases depending on the severity of the disease. It revealed that actual face-to-face treatment is more urgent than the user using the mobile app service. It can also be assumed that the user forgoes treatment for the companion animal due to cost problems. On the other hand, in mild cases, users are interested in both expert treatment or counseling and general counseling through collective intelligence among other pet owners. The H7 results reveal that among those who had negative perceptions about telemedicine for companion animals, those with a high degree of ease of use of mobile app services have a high proportion of positive evaluations about telemedicine services for people. This trend disproves the claim that people who find it easy to use mobile apps are more familiar with the IT environment than are those who do not.

**Table 8.** Result of moderation effect.

| Hypothesis | Path | t-Value (for H5a, b) /Z-score (H6, 7) | Test Result |
|---|---|---|---|
| H5a | Ease of use of mobile application service → negative perception of pet telemedicine service | −3.472 *** | Accept |
| H5b | Ease of use of mobile application service → negative perception of telemedicine service for people | −2.627 ** | Accept |
| H6 | Health condition → severity of disease → positive perception of pet telemedicine service | 8.427 *** | Accept |
| H7 | Negative perception of pet telemedicine service → Ease of use of mobile application service → negative perception of telemedicine service for people | 3.712 *** | Accept |

(Significant Level: ** $p < 0.05$, *** $p < 0.01$).

Despite the significance of this study, it has some limitations. First, assuming the widespread use of telemedicine for humans, advanced technologies such as X-ray and ultrasound cannot be applied. In this respect, simple comparisons with simple techniques such as pet telemedicine are problematic. This is similar to the same fundamental problem with safety that medical associations consistently highlight. Second, in pet telemedicine, the animal's unusual behavior or abnormal symptoms can be detected through dialogue with its owner, but the diagnoses may differ depending on the person expressing the symptoms. In other words, because pet owners who experienced the same symptoms when posting on social media or portal communities have different opinions and methods of expression,

remote prescriptions may also vary. Third, while conducting this study, we felt that the research on pet telemedicine platforms was insufficient. There are quite a few communities that serve as a forum for dialogue and communication, in addition to being a professional medical platform. In other words, although a specialized online medical device is required, we have not been able to clearly present a specific alternative for establishing a platform for such telemedicine.

The following is a summary of future research plans based on the problems identified. First, it is necessary to investigate research on telemedicine for humans. There is no actual commercialized domestic app for this, but a survey through a demo app can be conducted, based on which a perception survey on remote medical treatment for humans will be possible. Second, it is necessary to upload objective data in order to solve problems that are conveyed based on subjective judgment and different methods of expressing diseases. There is also a need for specialized devices with frames that can be accurately judged by medical personnel based on these data. With regard to "FITPET," the UI used in this study, the method of using the animal urine test kit is explained in detail in the app, which also provides clear standards for shooting and uploading photographs and videos. It also includes a function of recommending visits in some cases when abnormal signs are found in the photographs. The higher the number of such specialized telemedicine cases, the greater the possibility of telemedicine for humans and research for its development. This way, a continuous and concrete plan for the development of a professional telemedicine platform can be presented in the future.

## 6. Conclusions

This study makes several academic contributions and has practical implications for telemedicine. First, preceding studies confirmed that non-face-to-face treatment can be a good solution to medical problems in an environment in which it is not possible to easily visit hospitals or health centers due to the recent COVID-19 pandemic. In particular, as evidenced in studies from many major countries, patients with underlying diseases have more vulnerable immune systems, which could positively impact the introduction of telemedicine. Second, we conducted a survey among users of a commercialized mobile app service that provides pet telemedicine services and measured the adoption period of companion animals, their health status, ease of use of mobile apps, and positive and negative perceptions about telemedicine services to test the hypotheses. As a result of the statistical test, as explained by Tversky and Kahneman's prospect theory, users with negative perceptions about pet telemedicine were found to have significant negative perceptions about telemedicine for humans. However, in the case of people with positive perceptions, no significant conclusions were reached. This is generally consistent with the notion that people have a greater intensity of negative than positive perceptions. In other words, the concept transfer of negative recognition was much larger than that of positive recognition. Third, as explained by the additional hypotheses, people's perceptions about telemedicine differed according to the severity of their companion animal's disease. Substituting this with telemedicine for people, we can expect it will be much more effective to apply telemedicine to mild patients than to the severely ill. In addition, in light of the fact that the degree of ease of use of mobile apps acts as a moderating effect on the perceptions about telemedicine services, we can conclude that applying telemedicine to young people rather than middle-aged people will be quite useful in changing negative perceptions. Moreover, if we apply the recent AI-based medical apps to telemedicine, this will have a positive effect on the change in people's perceptions.

The practical implications of this study are as follows: First, it investigates the possibility of introducing telemedicine to humans by analyzing the pet telemedicine mobile app service. It can also have a positive effect on the future medical paradigm change and bring about a gradual improvement of the legal system. Second, it was found through comparative studies that the medical field's service app platforms are very poor compared to those of other industries, which we have highlighted as a problem. In fact, even a price

comparison app of a fairly well-known veterinary hospital reveals poor information in a specific region, reflecting the weak status of the platform to member companies. These platforms need constant improvement. Third, it has been found that remote medical care can attract high-quality services by investing a small amount of time and money for residents in blind spots where access to medical institutions is difficult. In particular, in mild cases, this effect can be maximized; future legalization of telemedicine should consider this aspect. Finally, we hope that this study could help solve the conflict between the government and medical associations over telemedicine.

**Author Contributions:** Conceptualization, Y.S.; data curation, S.H. and J.K.; formal analysis, S.H., Y.S., and J.K.; funding acquisition, J.K.; methodology, S.H. and J.K.; project administration, J.K.; resources, J.K.; software, S.H. and Y.S.; validation, Y.S.; writing—original draft preparation, S.H. and J.K.; writing—review and editing, J.K. All authors have read and agreed to the published version of the manuscript.

**Funding:** This research received no external funding.

**Conflicts of Interest:** The authors declare no conflict of interest.

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
