# Peer review of "Evaluation of AI-Assisted Telemedicine Service Using a Mobile Pet Application"

_applsci, doi:10.3390/app11062707_

Round 1

Reviewer 1 Report

Authors evaluate the users' acceptability of telemedicine through their acceptability to telemedicine systems for pets.

The main idea is very original, and it needs to overcome the limitation on telemedicine adoption that Korean government is currently addressing.

A statistical analysis based on a questionnaire has been conducted. The article is technically sound and the adopted methodology has been well described.

The main issue of this article is its length that is due to a very long related works section, that includes not necessary references, such as examples of app in different domains, a classification of AI based technology in medicine that is too general. Since the main topic of this article is telemedicine, related works section  should focus on other telemedicine systems that are used during this pandemic period (e.g. https://doi.org/10.1109/ISCC50000.2020.9219718 , https://doi.org/10.1109/JIOT.2021.3050775 ) and on other telemedicine evaluation works ( e.g. https://doi.org/10.1097/AOG.0b013e318224d110 , https://doi.org/10.1186/s12911-021-01407-y , https://dx.doi.org/10.12688%2Ff1000research.26794.1  )

Related works section is needed to highlight what are the limits of the state of the art and how the proposed methodology overcomes them.

In the following some minor comments:

  • Please do not use the term Chapter, but paragraph
  • Please add a reference/link to the apps that are mentioned at page 9, row 323
  • Please add a link to OnPanel website
  • Please add links to the softwares that have been used in the experimental part
  • Please proofread the article for typos

Reviewer 2 Report

Overview

The paper reports a survey on the use of AI and telemedicine services for different purposes. It also reports a study to predict the situation in which telemedicine services for humans will be introduced in the future and to suggest the possibility of treatment and service level.

The paper is well written and I find all the relevant ideas easy to understand and to follow.

The authors follow standard methodologies on the experimental part of the manuscript.

I suggest some minor corrections and adjustments of the manuscript as stated in my comments.

Comments (C)

C1 – Title of the paper. Since the paper is mainly a survey, I wonder if the authors can change the title of the paper from “Evaluation of AI-Assisted Telemedicine Service Using a Mobile Pet Application” to something that has the word “survey”  in the title; for instance, “Survey of Artificial Intelligence Assisted Telemedicine Services – a Mobile Pet Application”. Please also avoid the use of acronyms in the title.

C2 – Line 48.

for the outbreak of a new virus, such as the recent COVID-19

->

for the outbreak of a new virus, such as the recent SARS-CoV-2

C3 – Lines 48 and 49.

The COVID-19 situation has provided an o

->

The COVID-19 pandemic situation has provided an o

C4 – Line 54. Caption of Table 1.

We have “…in major countries.” This “major” countries classification is in respect to what criterion? Please state it clearly (in what sense is a major a country in this regard).

C5 – Lines 88 to 102.

When describing the paper organization, please use “sections” instead of “chapters”. The paper is organized into sections and not into chapters.

On the description of the remainder of the paper, please clearly state the contents of all the sections of the paper. There is no clear information regarding sections 5 and 6.

C6 – Line 88

Chapter 2 will review studies

->

Section 2 reviews studies

C7 – Line 90

Please define the IT acronym on its first appearance.

C8 – Line 96

In Chapter 3, we will perform

->

In Chapter 3, we perform

C9 – Line 99

In Chapter 4, we will evaluate

->

In Chapter 4, we evaluate

C10 – Line 164

to the recent COVID-19 can be said

->

to the recent COVID-19 outbreak can be said

C11 – Table 2. Please define the ICT acronym on its first appearance.

C12 – Line 192

We have “or strokes based on CT scans,”. Please define the CT acronym on its first appearance.

C13 – Line 193

Please define the MRI acronym on its first appearance.

C14 – Lines 197 and 198

The authors state that “it can be reproduced inexpensively across the world [30][31].” Notice that in some cases, there may be costs involved (eg. patents, copyright, and software purchase) and in these cases there are real costs. Please rephrase this sentence.

C15 – Section 2.2. Near line 185.

On the text, please refer to Figure 1. Please introduce the contents of this figure on the main text.

C16 – Line 234.

Please change “Figure 1 shows the increasing trend of” -> “Figure 2 shows the increasing trend of…”

C17 – Table 3 (and also Tables 2 and 1).

If possible, please add the vertical column separator. For instance, in Table 1 we have “glued text” on columns 2 and 3: “However, legalization or medical system cannot support the Salud Pública de”

On Table 3, we have “glued text” on columns 1 and 2: “medical institutionquired”

C18 – Line 318.

or mobile social network services They even issue

->

or mobile social network services. They even issue

C19 – Line 354.

Please add a proper literature reference to the partial least squares (PLS) technique. Please state also some other good techniques in the literature for this purpose.

C20 – Line 366. Caption of Table 5. On the caption of this table, please explain the meaning of the CR, AVE, and “CRB Alpha” columns.

C21 - On Table 5, the questions are not presented sequentially. Moreover, Q9, Q12, Q14, and Q24 are missing. Please revise this.

C22 – Line 399

H1:Positive perceptions

->

H1: Positive perceptions

C23 – Line 407

on the usability of technology, we

->

on the usability of the technology, we

C24 – Line 412

H3a:Positive perceptions

->

H3a: Positive perceptions

C25 – Line 433

has been with the use for a long time

->

has been with the user for a long time

C26 – Section 5 and section  6

Section 5 is rather small and it seems to be focused only on the results of Table 8. For a section with the discussion of the experimental results, I would expect that the contents of this sections would address all the experimental results (a recap of the experimental results).

Section 6 is rather long. I suggest that some parts of section 6 move to section 5. I also suggest to create subsection 6.1 – Future work.

C27 – References

Please revise the title of the paper in ref. 13. This title is entirely in capital letters.

Please revise ref. 57. It seems to be incomplete.

Reviewer 3 Report

The paper has the following flaws that need to be fixed:

1) The introduction section mentions chapters and instead the authors need to use the term sections

2) The conclusion section is too long. Many parts of this section needs to be moved to discussion

3) There is no need to discuss AI for heart diseases in section 2. Instead focus on the core subject matter in the paper.

Reviewer 4 Report

The authors investigated the possibility of telemedicine for humans through a mobile application targeting pets. The study is well designed, with solid analysis and plausible conclusions. The paper is relatively well organized. The plagiarism checker found no significant overlap with the base documents. Some comments are included below.

Major comments

  1. The research conducted is complex and the manuscript is relatively long. Therefore, I recommend that the authors give a brief description of the research steps at the beginning of the Introduction section. It might even be better to use a diagram.
  2. I suggest that the authors shorten or even omit the last paragraph in the Introduction section. Such a content announcement usually comes into play when writing a book. I suggest that this paragraph be replaced with an overview of the purposes and goals of the paper.
  3. The Discussion section desperately needs to be expanded and the Conclusions greatly shortened. The paragraphs and ideas presented in the Conclusions section should be moved to the Discussion. The conclusions should be short and concise. These changes will make an important contribution to the transparency and not least to the final quality of the article.

Minor comments

  1. SAS UE is an excellent software, but it is not open source (I point out the difference between open source and free software) (page 10, line 352).
  2. "Z-score" or "z-score" (page 10, line 361).
  3. Please explain and comment on abbreviations in tables throughout the manuscript.
  4. Provide an appropriate reference for the threshold of factor loadings (page 11, line 368).
  5. Explain to the reader what assumptions were made when conduction the t-tests in Table 4.
